



**Cultivated grasslands present a higher soil organic carbon sequestration efficiency under leguminous**
**than under gramineous species**
Yu Liu[a,b], Fu-Ping Tian[c], Peng-Yan Jia[a,b], Jing-Ge Zhang[a,b], Gao-Lin Wu[a,b]*
[a] State Key Laboratory of Soil Erosion and Dryland Farming on the Loess Plateau, Institute of Soil and Water Conserva
tion, Northwest AandF University, Yangling Shaanxi 712100, China;
[b] Institute of Soil and Water Conservation, Chinese Academy of Sciences and Ministry of Water Resources, Yangling
Shaanxi 712100, China;
[c] The Lanzhou Scientific Observation and Experiment Field Station of Ministry of Agriculture for Ecological System in the Loess
Plateau Area, Lanzhou Institute of Animal and Veterinary Pharmaceutics Sciences, Chinese Academy of Agricultural Sciences,
Lanzhou, Gansu 730050 China;
* Corresponding author e-mail address: gaolinwu@gmail.com (G.L. Wu)

12                                                    **ABSTRACT**

The establishment of grassland on abandoned cropland has been proposed as an effective method of
mitigating climate change by increasing soil organic carbon (SOC) storage. In this study, five cultivated
grasslands were established (three leguminous species -*Coronilla varia*, *Onobrychis viciaefolia*, *Medicago*
*sativa*, and two gramineous species-*Poa annua*, *Agropyron cristatum*), one uncultivated, one natural
grassland to examine how the SOC storage, sequestration rate and sequestration efficiency to change for 5
years restoration in semi-arid area. Our results showed that the cultivated leguminous grasslands had greater
total biomass, SOC storage, SOC sequestration rate and efficiency than gramineous grasslands. The greater
soil carbon (C) accumulation in leguminous grassland was mainly attributed to higher biomass production.
Leguminous grasslands accumulated more SOC than gramineous grasslands by 0.64 Mg C·ha$^{-1}$·yr$^{-1}$. The
average SOC sequestration efficiency in leguminous grassland (1.00) was about 2 times greater than
gramineous grassland (0.34). The results indicate that cultivated leguminous grasslands sequestered more
SOC with higher SOC sequestration efficiency than cultivated gramineous grasslands in arid and semi-arid
areas.
**KEY-WORDS:** Cultivated grassland, Carbon sequestration, Gramineous, Leguminous, SOC





## 1   Introduction


The soil is a key component of the Earth System and contribute to services, goods and resources to the
humankind (Brevik et al., 2015). Soil stored more carbon (C) than the atmosphere and vegetation (Köchy et
al., 2015; Keesstra et al., 2016). Soil organic carbon (SOC) as a key component of the global carbon cycle
and its potential to sink from atmosphere carbon dioxide ($CO_2$) have been widely discussed in the scientific
literatures throughout the world (Guo and Gifford, 2002; Lal, 2004; De Deyn et al., 2008; Deng et al., 2014a;
Parras-Alcántara et al., 2015). Thus during recent decades, massive emphasis had been given in SOC storage
and sequestration on global scale. In the terrestrial ecosystem SOC pool dynamics were affected by many
factors, such as climate change (Lal, 2004; Field et al., 2007), management practices (Luo et al., 2010; Ono
et al., 2015), land use etc. (Post et al., 2000; Don et al., 2011; Deng et al., 2014b; Muñoz-Rojas et al., 2015).

SOC plays an extremely important role in control of soil fertility and cropping system productivity and

sustainability (Hurisso et al., 2013; De Moraes Sá et al., 2015), particularly in low-productivity arid and
semiarid agro-ecosystems (Behera et al., 2015). To develop farming methods that conserve SOC is therefore
of a great importance (Lal, 2004). Cultivated grassland has much more advantages than natural grassland
regeneration, such as accelerating vegetation restoration and improving grassland productivity. Establishing
artificial grassland is one type of land uses to restore vegetation and improve SOC (Fu et al., 2010; Li et al.,
2014; Wu et al., 2010). In grassland, atmosphere carbon was sequestrated through photosynthesis and
respiration, then carbon fixing in stable SOC pool or releasing back into the atmosphere (Post et al., 2000).
Therefore, studying the carbon sequestration in grassland ecosystems can help to identify the magnitude of
global carbon sinks and sources (Li et al., 2014).

The balance of Soil carbon pool is determined by the carbon input from leaf and root and its

mineralization in soil, and output in decomposition processes of soil organic matter by soil microbes and
respiration from plant roots (Amundson, 2001; Garcia-Diaz et al., 2016). The biomass fraction resulting in



SOC build-up (plant residuals) was strongly affected by management practices including the selection of
plant species (Don et al., 2011). Species composition had a great role in determining the aboveground
productivity (Liu et al., 2016). Over relatively long time, the proportion of the aboveground biomass enters
soil as organic matter and incorporates into soil through physical and biological processes. For example,
some leachates from plant material in the litter layer, root exudates, solid decomposed litter and fragmented
plant structure materials (Jones and Donnelly, 2004; Novara et al., 2015). The amount of plant residuals
returned to the soil directly affected the SOC (Musinguzi et al., 2015; Wasak et al., 2015), and mostly
perennial plants were managed with high planting densities to produce greater biomass exports (Hobbie et al.,
2007; Köchy et al., 2015).

Vegetation degradation and exponential population growth have caused massive amounts of soil and

water to be lost. The Chinese government has implemented the most ambitious ecological program titled
'Grain-for-Green' Project (converting degraded, marginal land and cropland into grassland, shrubland and
forest), with the objective of transforming the low-yield slope cropland into grassland, reducing soil erosion,
maintaining land productivity and improving environmental quality (Fu, 1989; Liu et al., 2008;). The large
scale of the project indeed enhanceed carbon sequestration capacity in China, especially in arid and semi-arid
areas (Chang et al., 2011; Song et al., 2014).

Many prior studies about SOC have paid much attention to conversion from farmland to grassland,

shrubland or forest (Fu et al., 2010; Deng et al., 2014a). The main dominant grass species used in the project
are leguminous and gramineous (Jia et al., 2012; Wang et al., 2015). However, less attention has been
devoted to the SOC among different plant species grasslands. In current study, we have focused on
ascertaining the influence of leguminous and gramineous grasslands on SOC sequestration capacity and
efficiency. Many studies had demonstrated that there is a significant and positive relationship on SOC and
nitrogen (Deng et al., 2013; Zhu et al., 2014). So we hypothesize that the leguminous grassland has the



higher SOC sequestration capacity than gramineous grassland. More specifically, our objectives are: (i) to
analyze the effects of SOC stock and sequestration under different grasslands; (ii) to determine which type of
cultivated grassland might better improve SOC storage in arid and semi-arid areas.
**2 Material and methods**
**2.1 Experimental site and design**
The study site was located in Gongjiawan County (103°44′ E, 36°02′ N, 1966 m a.s.l.) of Lanzhou, Gansu
Province, China. The site is the semi-arid continental temperate monsoon climate zone. The data from the
National Meteorological Information Center of China showed that the mean annual temperature was 9.3 ℃
(2008-2012), and the minimum and maximum values were -23.1 ℃ and 39.1 ℃ (2008-2012), respectively.
The annual cumulative temperature above 10 ℃ was between 1900 and 2300 ℃·d, and above 0 ℃ it was
3700 ℃·d. The mean annual precipitation was 324.5 mm, and which approximately 80% falls during the
growing season (from May to September). The topography of study area was typical characteristics of the
Loess Plateau, such as plains, ridges and mounds, etc. The elevation of study site was about 1700 m. The
main soil type was Sierozem, which is a calcareous soil and characteristics of the Chinese loess region (Li et
al., 2010). Sierozem is the soil developed in the dry climate and desert steppe in warm temperate zone, which
has low humus and weak leaching (National soil census office, 1998). There is the patch or pseudohyphae
calcium carbonate deposition and strong lime reaction within full sierozem profile (Shi, 2013).

The experimental site was originally under sorghum (*Sorghum bicolor* L.) continuously from 1970 to

2005 and was abandoned from 2005 to 2008. In 2008, five cultivated grasslands, one uncultivated grassland
(abandoned cropland, Un-G), one natural grassland (Na-G) were established in the study site. Five main
forage grasses, widely grown across in semi-arid areas, were selected to establish five types of cultivated
grassland, namely three leguminous species (*Coronilla varia* L., L-CV; *Onobrychis viciaefolia* Scop, L-OV;
*Medicago sativa* L., L-MS) and two gramineous species (*Poa annua* L., G-PA; *Agropyron cristatum* L.



Gaertn., G-AC) (Table 1). Three experimental plots 10 m $\times$ 20 m were established randomly within each of
the grassland areas. The forage grasses were planted in early April of 2008, and all plots were weeded
manually and watered three times (April, June, October) annually from 2008 to 2012 to preserve the
monocultures. The plots did not fertilized during cultivation. All the plots were harvested once a year in
October.

**2.2 Aboveground plant and belowground biomass sampling**

Aboveground biomass was measured by harvesting the upper plant parts, by clipping their stems at the soil
surface, from ten quadrats (1 m $\times$ 1 m) in each plot randomly in late August every year (2008-2012). All
green aboveground plant parts were collected separately by each individual species, and all the litter layer
also were collected with the labeled envelops. Then these samples were dried at 105 ℃ until their mass was
constant, and then their mass was weighed and recorded.
Belowground biomasses and soil samples were taken in the four corners and the center of the quadrats
where were the aboveground biomass sampling points. Belowground biomass were collected using a soil
drilling sampler with 9 cm inner diameter from 0-100 cm soil layer, and separated into increments every 10
cm. The roots in the soil samples were obtained by a 2 mm sieve. Then the remaining roots in the soil
samples were isolated by shallow trays, and allowing the flowing water from the trays to pass through a 0.5
mm mesh sieve. All the roots samples were oven-dried at 65 ℃ then weighed.

**2.3 Soil sampling and determination**

In each quadrat, the same layer samples were mixed together and be composed of a composite sample. The
samples were passed through a 2-mm sieve to remove the roots and other debris. A 5 cm diameter and 5 cm
high stainless steel cutting ring (~100 cm$^3$) was used to measure soil bulk density (BD) at adjacent points to
the soil sampling quadrats. Soil bulk density was measured at the depth of 0-100 cm. The dry mass were
measured after oven-drying at 105 ℃. Soil organic carbon content was measured using the method of the





vitriol acid-potassium dichromate oxidation (Walkley and Black, 1934). All the analyses of one sample were
carried out in three replications.
**2.4 Relative calculation**
BD was calculated depending on the oven dried weight of the composite soil samples (Deng et al., 2013).
The SOC stock for each soil layer was calculated using the equation as follows (Deng et al., 2013):
$C_s = BD \times SOC \times D/10$ (1)
where, $C_s$ is the SOC stock (Mg ha$^{-1}$); BD is the soil bulk density (g cm$^{-3}$); SOC is the soil organic carbon
content (g kg$^{-1}$); and D is the thickness of the sampled soil layer (cm).
The SOC sequestration rate (SSR, Mg ha$^{-1}$ yr$^{-1}$) was calculated as follows (Hua et al., 2014):
$SSR = (C_t - C_0)/t$ (2)
where, $(C_t - C_0)$ is SOC sequestration; $C_t$ is the SOC stock in 2012; $C_0$ is the SOC stock in 2008; t was
the duration of experiment.
The SOC sequestration efficiency was estimated using the SOC sequestration in the weight of total
biomass (aboveground biomass and belowground biomass) of per unit area:
$C_{se} = \triangle C / B_T/10$ (3)
where, $C_{se}$ is the SOC sequestration efficiency; $\triangle C$ (Mg ha$^{-1}$) is the SOC sequestration from 2008 to
2012; $B_T$ (kg m$^{-2}$) is the total biomass (above ground and below ground) from 2008 to 2012.
**2.5 Statistical analyses**
The data were examined for normality by the Shapiro-Wilk test and homogeneity of variances by the Levene
test before analysis. To get a normal distribution, performing statistical tests not normally distributed data
were log-transformed. All data were expressed as mean values ± standard error (M ± SE). The means of SOC
sequestration rate and SOC sequestration efficiency among the different grassland types were assessed using
One-way Analysis of Variance (ANOVA). Two-way ANOVA of Type III was performed to test the influences




of grassland types and time on SOC content, storage and bulk density. Tukey test was conducted to test the
significance at $p < 0.05$ level. All the statistical analysis was performed with SPSS version 18.0 (SPSS Inc.,
Chicago, IL, USA).
**3 Results**
**3.1 Aboveground net primary productivity**
Between 2008 and 2012, the five cultivated grasslands in general had greater total biomass values than the
uncultivated grassland and natural grassland (mean by 189.36%). In addition, the three grasslands cultivated
with the leguminous species had greater annual total biomass than the two gramineous grasslands (mean by
72.6%), which lead to a greater total biomass values of the three leguminous species at the end of the study
period. In particular, the L-MS grassland consistently had the greatest total biomass throughout the study
period (Fig.1a).
**3.2 Soil SOC content and controls**
Results from two-way ANOVA showed that the plots types, year and interactions all significantly affected
total biomass, SOC content, and BD (Table 5). The average SOC content followed leguminous grassland >
natural grassland > uncultivated grassland > gramineous grassland, and it increased over time in all
grasslands (Table 2). The L-MS grassland had the highest SOC content among the grasslands during the
study period. The effects of grassland type on soil bulk density followed uncultivated and natural grassland >
gramineous grassland > leguminous grassland (Table 3).
**3.3 Soil organic carbon stock change**
The SOC storage under all the grasslands increased significantly throughout the study period (Table 4), with
the three cultivated leguminous grasslands further significantly greater than those under the two gramineous
grasslands. To be specific, in the 0-20 cm soil layer, the SOC storage under the L-MS, L-CV and L-OV
grasslands increased from 9.73, 5.20, 7.27 Mg C ha$^{-1}$ to 14.95, 13.54, 12.05 Mg C ha$^{-1}$, respectively, during





the experimental period.
**3.4 Soil carbon sequestration rate and sequestration efficiency**
SOC sequestrations in three leguminous grasslands were greater than two gramineous grasslands (mean
by196.74%; Fig.1c). Three leguminous grasslands accumulated C with an average rate 1.00 Mg C·ha$^{-1}$·yr$^{-1}$
which is more than the 0.34 Mg C ·ha$^{-1}$·yr$^{-1}$ in gramineous grassland, and more than the average of
uncultivated and natural grasslands (0.25 Mg C ·ha$^{-1}$·yr$^{-1}$).
The mean SOC sequestration efficiency in the leguminous grassland was about 0.26, which was
significantly greater than others grassland types ($p<0.05$; Fig.1d). The maximum and minimum efficiency
values were 0.37, 0.08 in L-CV, G-PA grassland, respectively. The average SOC sequestration efficiency in
leguminous grassland was two times greater than gramineous grassland.
**4 Discussion**
SOC content of all grassland plots showed some differences between each other (Table 2 and 3). The average
SOC content in leguminous grasslands was 2.64 g kg$^{-1}$ and that in gramineous grasslands was 1.97 g kg$^{-1}$.
Moreover, both soil bulk density of leguminous and gramineous grasslands were 1.46 g cm$^{-3}$ in 2008. The
reasons for the SOC content difference result from precedent soil conditions and cultivated grasses. Different
types of cultivated grasses, as well as the precedent soil conditions are probably the two reasons for the SOC
content differences between leguminous and gramineous grasslands. The irregular distribution of precedent
plant residues and roots resulted in the patch of nutrients in the soil and changing the soil physical conditions,
such as SOC and BD. In addition, mutualistic symbionts (N-fixing bacteria and mycorrhizal fungi) are also
an important source of carbon input to soil, especially in actively growing plants (Bardgett et al., 2005).
Symbiosis can increase plant productivity through enhanced the acquisition of limited resources. Moreover,
mycorrhizal fungi can immobilize carbon in their mycelium and improve carbon sequestration in soil
aggregates (Rillig and Mummey, 2006). Our results demonstrated that a key variable associated with higher



SOC content in leguminous grasslands than gramineous grasslands is the greater total biomass accumulation.
The leguminous grasslands had both higher above- and belowground biomasses than gramineous grasslands.
Total biomass was 16.35 kg m$^{-2}$ in leguminous grasslands, which is 9.47 kg m$^{-2}$ more than gramineous
grasslands from 2008 to 2012. In addition, the grasslands in our study without grazing and only harvesting
the aboveground biomass annually, so all the aboveground stubble and plant litters be input to soil as a
carbon supply. SOC mostly originates from decaying this aboveground and belowground plant tissue, so
greater soil C accumulation was mainly ascribed to increasing soil C input from higher biomass production
(Deng et al., 2014c; Wu et al., 2016). Previous studies had showed that plant regulated SOC stock by
controlling carbon assimilation, its transfer and storage in plant root system, then through plant respiration
and leaching its release from soil to atmosphere(De Deyn et al., 2008). Deng et al. (2014c) have found that
plant biomass is the key driver in soil carbon sequestration. In this study, the SOC increased dramatically in
leguminous grassland due to the greater total biomasses of the leguminous grasses, and the increased soil
carbon inputs from the litter layer and root biomass (De Deyn et al., 2008; Wu et al., 2010; Novara et al.,

2015).

SOC sequestration rates in the cultivated leguminous grasslands were significantly higher than that in the
gramineous grasslands (Fig. 1c). This maybe resulted from SOC sequestration and the different
decomposition rates in soils, because the cultivated leguminous and gramineous grass species result in
multifarious nutrient conditions. The slower rates of decomposition might make soil carbon storages
increased faster in more nutrient-poor soils (Vesterda et al., 2002; Deng et al., 2014a). L-CV grassland has
the highest SOC sequestration rate and efficiency but with the lowest total biomass among the leguminous
grasslands. The reasons maybe the different species with the various C sequestrate capability, but the
potential mechanism under each species need further studies to demonstrate. Leguminous grasslands
achieved greater SOC sequestration rates due to the total biomass was higher than that in the gramineous





grasslands. Litter and fragmented plant parts at the soil surface are decomposed by micro-organisms and are
gradually incorporated into the soil through some complex processes (Novara et al., 2015). Legumes had the
ability to develop root nodules and to fix nitrogen in symbiosis with compatible rhizobia, which should
improve the soil nutrient status. Moreover, many previous studies had demonstrated that soil carbon and total
nitrogen are significantly and positively correlated (Deng et al., 2013; De Oliveira et al., 2015). Therefore, it
might be expected that the cultivated leguminous grasslands had significantly improved soil N contents that
led to a greater carbon sequestration ability than the non-leguminous grasslands. Furthermore, the resulting
increase in fertility of the soils under the leguminous grasses should facilitate the increased productivity of
the plants. Our results showed that SOC sequestration efficiency under leguminous grasslands was evidently
greater than that in the gramineous grasslands (Fig. 1d). It is noteworthy that L-MS grassland had the highest
total biomass 22.59 kg m$^{-2}$ which is 2.38 times as much as the average of gramineous grasslands (Fig.1a),
moreover, SOC sequestration in L-MS grassland is 3 times as much as the average of gramineous grasslands
(Fig.1b). So the SOC sequestration efficiency in L-MS grassland is higher than gramineous grasslands.
**5 Conclusion**
Leguminous grasslands had greater SOC storage, sequestration rate and efficiency than gramineous
grasslands. The greater soil C accumulation of leguminous grasslands was mainly ascribed to higher biomass
production. Leguminous grasslands accumulated an average rate of 0.64 Mg C·ha$^{-1}$·yr$^{-1}$ more than
gramineous grasslands. The average SOC sequestration efficiency in leguminous grasslands was 2 times
greater than that in the gramineous grasslands. The results indicate that cultivated leguminous grasslands
sequestered more soil carbon with a higher SOC sequestration efficiency than cultivated gramineous
grasslands in arid and semi-arid areas.
**6 Acknowledgement**
We thank the editor for suggestions on this manuscript. This research was funded by the National Natural





Science Foundation of China (41371282, 41525003, 31372368, 41303062), the Youth Innovation Promotion
Association CAS (2011288), the "Light of West China" Program of CAS (XAB2015A04), Lanzhou Institute
of Animal and Veterinary Pharmaceutics Sciences of Chinese Academy of Agricultural Sciences
(CAAS-ASTIP-2014-LIHPS-08).

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



**Table 1.**    Description of studied grassland types.

| Glassland types | Species | Seeding rates (kg ha$^{-1}$) |
|---|---|---|
| Leguminous grassland | *Coronilla varia* L. | 7.5 |
| | *Onobrychis viciaefolia* Scop. | 30 |
| | *Medicago sativa* L. | 12 |
| Gramineous grassland | *Poa annua* L. | 7.5 |
| | *Agropyron cristatum* (L.) Gaertn. | 15 |
| Uncultivated grassland | Abandoned cropland. Natural successional species were present, e.g., *Chenopodium album L.*, *Agropyron cristatum* L. | |
| Natural grassland | A local native grassland community. Dominant species were *Stipa breviflora* Griseb., *Stipa aliena* Keng, *Artemisia capillaris* Thunb., *Artemisia annua* L. | |


**Table 2.**  Soil C concentration (M ± SE g kg$^{-1}$) in different years and grassland types. Note: The grassland
types were: L-Cv, *Coronilla varia*; L-Ov, *Onobrychis viciaefolia*; L-Ms, *Medicago sativa*; G-Pa, *Poa annua*;
G-Ac, *Agropyron cristatum*; Un-G, uncultivated grassland; Na-G, natural grassland. Values followed by
different lower-case letters within columns and upper-case letters within rows are significantly different at
$p < 0.05$.

| Grassland types | 2008 | 2009 | 2010 | 2011 | 2012 |
|---|---|---|---|---|---|
| L-CV | 2.31±0.04dE | 3.09±0.05cD | 4.22±0.04bC | 4.91±0.02bB | 5.92±0.05bA |
| L-OV | 2.70±0.04bE | 3.33±0.02bD | 3.96±0.02cC | 4.69±0.08cB | 5.44±0.12cA |
| L-MS | 2.92±0.06aE | 3.62±0.05aD | 4.38±0.02aC | 5.55±0.09aB | 6.13±0.05aA |
| G-AC | 1.90±0.01gE | 2.13±0.03fD | 2.56±0.04eC | 2.94±0.03eB | 3.46±0.06dA |
| G-PA | 2.03±0.01fE | 2.14±0.02fD | 2.26±0.02fC | 2.57±0.01fB | 2.65±0.02fA |
| Un-G | 2.20±0.08eCD | 2.35±0.02eC | 2.42±0.04efC | 2.81±0.01eB | 3.16±0.02eA |
| Na-G | 2.53±0.08cB | 2.71±0.10dB | 2.80±0.12dB | 3.18±0.13dA | 3.26±0.06eA |








**Table 3.**  Soil bulk density (M ± SE g cm$^{-3}$) in different years and grassland types. Note: The grassland types

were: L-Cv, *Coronilla varia*; L-Ov, *Onobrychis viciaefolia*; L-Ms, *Medicago sativa*; G-Pa, *Poa annua*; G-Ac,

*Agropyron cristatum*; Un-G, uncultivated grassland; Na-G, natural grassland. Values followed by different

lower-case letters within columns and upper-case letters within rows are significantly different at $p<0.05$.

| Grassland types | 2008 | 2009 | 2010 | 2011 | 2012 |
|---|---|---|---|---|---|
| L-CV | 1.41±0.01dAB | 1.42±0.01bA | 1.39±0.01cB | 1.37±0.01cdC | 1.35±0.01dD |
| L-OV | 1.51±0.01aA | 1.46±0.01aB | 1.40±0.01bcdC | 1.36±0.01dD | 1.33±0.01eE |
| L-MS | 1.47±0.15bcA | 1.47±0.01aA | 1.43±0.02abAB | 1.39±0.02bcBC | 1.36±0.01cC |
| G-AC | 1.45±0.01cA | 1.46±0.02aA | 1.46±0.01aA | 1.45±0.01aA | 1.39±0.01bB |
| G-PA | 1.47±0.01bcA | 1.46±0.01aA | 1.39±0.01cB | 1.38±0.01cdC | 1.34±0.01eD |
| Un-G | 1.48±0.01bA | 1.47±0.01aB | 1.43±0.01abC | 1.42±0.01bD | 1.40±0.01aE |
| Na-G | 1.49±0.01abA | 1.48±0.01aB | 1.42±0.01bcC | 1.42±0.01bCD | 1.41±0.01aD |




**Table 4.** SOC stock (M ±SE Mg ha$^{-1}$) at the depth of 0-100 cm in different years and grassland types. Note:

The grassland types were: L-Cv, *Coronilla varia*; L-Ov, *Onobrychis viciaefolia*; L-Ms, *Medicago sativa*;

G-Pa, *Poa annua*; G-Ac, *Agropyron cristatum*; Un-G, uncultivated grassland; Na-G, natural grassland. Values

followed by different lower-case letters within columns and upper-case letters within rows are significantly

different at $p<0.05$.

| Grassland types | 2008 | 2009 | 2010 | 2011 | 2012 |
|---|---|---|---|---|---|
| L-CV | 31.49±0.31dE | 43.10±0.60cD | 57.92±0.87abC | 66.67±0.17bB | 79.34±0.80bA |
| L-OV | 40.05±0.36bB | 48.57±0.41bAB | 55.41±0.41bAB | 63.89±1.09bAB | 72.66±1.38cA |
| L-MS | 43.75±0.87aE | 53.69±0.89aD | 63.20±1.28aC | 77.50±1.62aB | 83.77±0.76aA |
| G-AC | 27.11±0.27fE | 30.87±0.60fD | 37.10±0.60cC | 42.53±0.33cB | 48.10±0.82dA |
| G-PA | 29.29±0.06eC | 30.80±0.36fB | 31.35±0.19dB | 35.38±0.06eA | 35.36±0.37fA |
| Un-G | 32.03±0.65dD | 33.83±0.18eC | 33.83±0.52cC | 38.72±0.17dB | 43.25±0.22eA |
| Na-G | 36.25±0.61cB | 38.40±1.25dB | 39.26±1.61cB | 44.74±2.00cA | 45.20±0.98eA |





**Table 5** Two-way ANOVA F and p values for the effects of plot types, year, and interactions on total biomass
(TB), soil organic carbon content (SOC), soil C storage, and soil bulk density (BD). Bold numbers indicate
statistical significance.

| Factor | df | TB | | SOC | | C storage | | BD | |
|---|---|---|---|---|---|---|---|---|---|
| | | F | *p* | F | *p* | F | *p* | F | *p* |
| **Grassland types** | 6 | 296.19 | <0.001 | 40.52 | <0.001 | 42.03 | <0.001 | 42.48 | <0.001 |
| **Year** | 4 | 100.67 | <0.001 | 49.37 | <0.001 | 41.05 | <0.001 | 7.24 | <0.001 |
| **interaction** | 24 | 32.57 | <0.001 | 2.30 | 0.001 | 2.00 | 0.001 | 7.36 | <0.001 |







**Figure captions**
**Fig. 1** Total biomass (a), SOC sequestration (b), SOC sequestration rate (c) and SOC sequestration efficiency
(d) for different grassland from 2008 to 2012,. Note: The grassland types were: L-Cv, *Coronilla varia*; L-Ov,
*Onobrychis viciaefolia*; L-Ms, *Medicago sativa*; G-Pa, *Poa annua*; G-Ac, *Agropyron cristatum*; Un-G,
uncultivated grassland; Na-G, natural grassland. Bars indicate mean ± standard error. Bars with the different
lowercase letter above them indicate there was significant difference between the means at *p*<0.05 level. The
dotted lines indicate the means of the same grassland types.

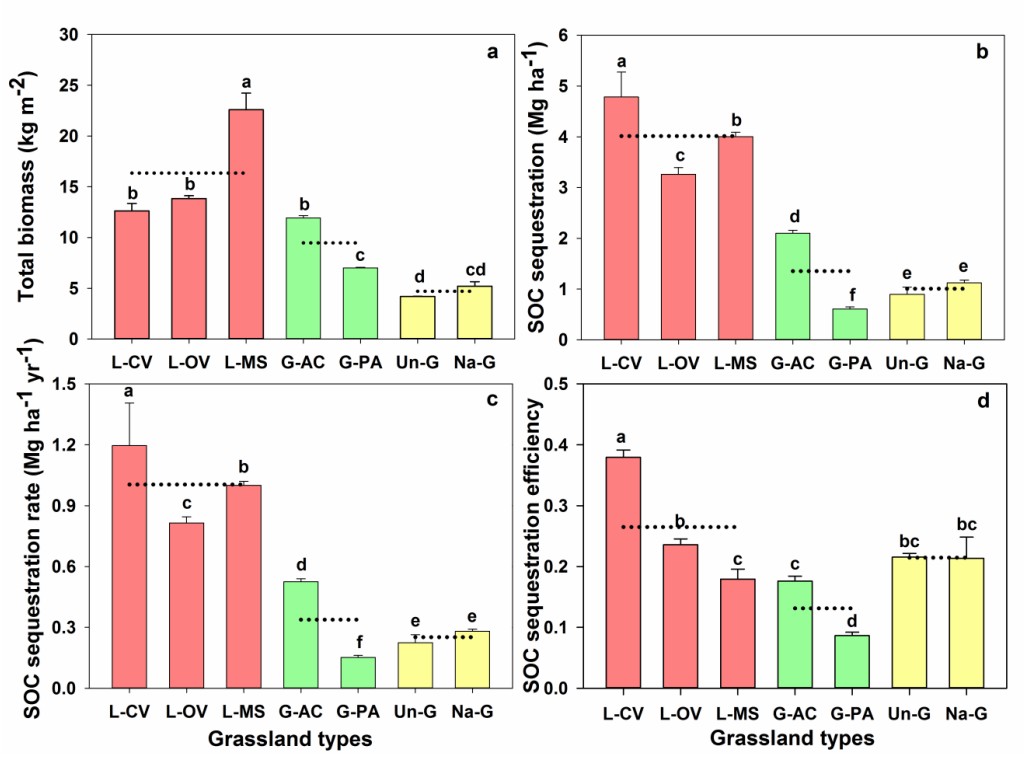
