# Peer review of "Cultivated grasslands present a higher soil organic carbon sequestration efficiency under leguminous than under gramineous species"

_Solid Earth, 2016_

## Referee Comment (RC1) · M. Ledevin (Referee) · 26 Sep 2016

Liu et al. propose a manuscript dedicated to the organic carbon sequestration under various grassland types. The topic is of interest with respect to the 'Grain-for-Green' Project sustained by the Chinese government. The prime interest of the paper is to decipher the optimal setting under which the transformation to grassland would permit the most efficient SOC sequestration in arid and semi-arid areas. In that regard, their results show the higher efficiency of leguminous species to retain carbon in soils, while gramineous species are much less effective.

The scope of the study and the main result are somehow clear, but the writing, however, is rather complicated and in parts lengthy...I strongly suggest a thorough rewriting of

the paper to achieve a more logical organization of the discussion, the removal of repetitions and the synthesis of lengthy parts such as the introduction, the presentation of the studied area and the presentation of the data.

More critical is the lack of rigor of the discussion. Many shortcuts and omissions renders the discussion very difficult to follow, and sometimes borderline when it comes to attribute a process to the observed data...the authors have to discuss more deeply their ideas.

Another key problem is that the article fails to attract attention on the importance of such work. Emphasis should be put on the importance of such contribution, not only in the introduction but also through the discussion itself: why does it matter? what's next? what are the perspectives? what broader applicability? etc. Without such perspective, it is hard to evaluate whether the study is of true interest or not for the topic.

Therefore, because of the poor organization, poor english writing, lack of perspective and lack of rigor of the discussion, I do not recommend this manuscript for publication. Instead, I suggest a deep correction before re-submission.

In the attached file, I provide more precise comments and suggestions to improve the manuscript. Because of the abundance of English mistakes, I did not include specific corrections, but instead strongly recommend some collaboration with a native english speaker to improve the writing.

Please also note the supplement to this comment:
http://www.solid-earth-discuss.net/se-2016-109/se-2016-109-RC1-supplement.pdf
* * *
[Figure]

**Supplement:**

The following points highlight the major problems I found in this manuscript and provides some ideas to enrich the discussion. I also propose a new organization for the discussion, which I hope will help the authors to find a more logical way of discussing their results.

1) **Many repetitions** can be avoided thorough the entire manuscript: rephrasing the same idea is not useful for the comprehension; it just adds length to the manuscript.
One telling example is the two juxtaposed sentences found Line 179 to 181:
Sentence 1: "The reasons for the SOC content difference result from precedent soil conditions and cultivated grasses."
Sentence 2: "Different types of cultivated grasses, as well as the precedent soil conditions are probably the two reasons for the SOC content differences between leguminous and gramineous grasslands."

2) **The description of the results** can be clarified by removing the sub-sections and preventing, again, many repetitions (e.g. Line 167-170, Line 171-174, etc). In fact, the results for each measured criteria (i.e. total biomass, SOC content, SOC storage, sequestration rate and efficiency) can be described in no more than 1 or 2 sentences. The sub-division is therefore unnecessary, more so because the titles themselves are not adequate.

- For example, the Section 3.1 is labeled "Aboveground net primary productivity" but the following text only describes the "total biomass". What are you actually talking about? The sum of above and belowground biomass? What do you call "net productivity"? You have to be consistent and specific about your terminology.
- The Section 3.2 is labeled "Soil SOC content and controls": the term soil is already comprised in the acronym SOC, and what are those controls you are referring to?

If you wish to preserve the sub-division of the results, use simple and accurate terminologies similar to the one you are referring to in the following text.
Note also that the first lines of the discussion (Line 176-178) belong to the results!

3) **The implications of the results** are not enough emphasized in the manuscript. Why does it matter? What perspective? This is somehow mentioned in the introduction, but you should recall this in your discussion and use it to put your work into the broader picture! This would contribute to make the study more attractive and significant for the scientific community.

4) **The abstract** lacks the key elements presented in the discussion, which are the importance of the micro-organisms and their effect on the degradation of plant tissues and nutrient availability, which in turn affects the SOC storage. The abstract should emphasize this, instead of only the higher efficiency of leguminous grasslands…also a word on the importance of the results would be welcome.

5) **The poor English writing**, such as grammatical mistakes, wrong choice of past and present tense conjugations, wrong verbs and misuse of vocabulary, renders difficult the comprehension and reading of the manuscript…I strongly recommend collaborating with native English speakers to thoroughly correct the paper!

6) **The discussion** is problematic in that it is poorly organized and lack scientific rigor. It looks like a succession of your results juxtaposed with ideas/results from other studies, but it badly misses the link between the two (to much shortcuts…), which can be avoided by providing an actual discussion of the processes behind. Some of the processes are partially

discussed, but the overall organization makes it hard to understand as you jump back and forth from one concept to the other.

More importantly, you have to discuss the differences between natural, cultivated and uncultivated outcrops, not only the difference between leguminous and gramineous!! Otherwise it is not necessary to present these results.

Similarly, most of the discussion seems dedicated to leguminous grasslands. You should also discuss why the gramineous grasslands are less efficient…

I strongly suggest reorganizing the whole discussion! For example, 1 paragraph per important idea…with a deeper discussion of each of the processes. To help the authors, here is a suggestion for re-organizing the discussion:

**Paragraph 1: Recall here the main goal of the study.**

Recall the reader here that you investigate the efficiency of SOC storage and especially compare the differences between gramineous and leguminous grasslands (recall briefly why it is important! Re-use key parts of the introduction for example).

Then, explain that your results show significant differences among the SOC content and storage efficiency between the various plots, which you believe is due to (1) the condition of the soil and (2) the nature of cultivated grasses. In the following discussion, you will explore and discuss various parameters that explain the variable efficiency in your samples.

**Paragraph 2: Basic principle of SOC regulation by plant**

First of all, you have to summarize the main role of plants in the local C cycle, based on the literature! This is important to further discuss the processes linked to this cycle!

Outputs of C from soils = Regulation by plants through carbon assimilation, transfer and storage in the root system, then through plant respiration and release to atmosphere as $CO_2$ = output from the soil!! Not sequestration!

Inputs of C to soils = The storage occurs only during the decay/decomposition of the above and belowground plant tissues (e.g. roots, leaves, etc).

**Paragraph 3: Important finding of your study**

You results show that leguminous grasslands are more efficient to store carbon: they show higher SOC content and higher rate of storage than gramineous grasslands. Moreover, they contain higher biomass, which may be the key of their efficiency... In the following paragraphs, you will try to explain what parameters can explain such difference between the leguminous and gramineous species.

**Paragraph 4: First parameter(s) = Soil conditions**

From your discussion, one key parameter seems to be the conditions of the soil. Here, you should explain in which aspects the soil can contribute significantly to the storage of carbon. For example, you mentioned that the irregular distribution of precedent plant residues and roots results in the irregular distribution of nutrients. It also affects the SOC and BD measurements. You have to discuss the significance of such "memory effect" from previous plantations! How does it affect your results? Can it account for the different SOC content between your outcrops? Certainly not, and you will look into other, more significant parameters in the following paragraphs.

**Paragraph 5: Second parameter(s) = capacity to incorporate C**

The second parameter is more focused on the species themselves: different species may incorporate more or less carbon according to their specific metabolism (cite other studies

here). Accordingly, the higher SOC content in leguminous grasslands may reveal a greater capability to incorporate C in their root system than gramineous species. Check in the literature if this is true: can you find actual numbers to show the different incorporation capacity of each species? Is it sufficient to explain your data?

**Paragraph 6: Third parameter(s) = Plant productivity**
Plant productivity seems to be the most important factor: the more productive the plants are, the more biomass they produce! Therefore, they integrate more carbon, which will be ultimately released during their decomposition (shown by other studies).
According to your results, the very high SOC content of leguminous grasslands is tightly linked to their higher biomass content, which may indicate greater productivity than gramineous grasslands: they produce more biomass and the corresponding high C content is transmitted to the soil during plant decomposition. Can you compare your results with other studies that found similar differences?

**Paragraph 7: Origin of greater leguminous productivity**
Then it is important to discuss why you think leguminous grasses are more productive!
One key idea is the capability of leguminous species to fix nitrogen within root nodules, with two consequences: (1) it promotes the fertility of the soil, which facilitates plant productivity, and (2) it promotes the symbiosis with micro-organisms, which in turns help with the fixation of limited nutrient, therefore enhancing the plant productivity. Both effects result in greater biomass production, and therefore elevated C content in the end.
Moreover, such micro-organisms are responsible for the decomposition of the plants, and therefore constitute the key of the transmission of the stored carbon into the soil! You say that the incorporation in the soil occurs through more complex processes: which ones?? No need to describe, but at least cite them!
Finally, micro-organisms may contribute themselves to the final SOC content by incorporating C in their mycelium: to what extent? Is it significant when you compare with the C brought by the plants themselves?
You also have to discuss the case of the gramineous! What kind of micro-organisms? Are the symbionts restricted to the leguminous samples? Why is the gramineous productivity lower? Etc.
Compare with other studies from the literature to confirm this idea.

**Paragraph 8: Sequestration rate**
Perhaps you can include here a paragraph discussing the sequestration rate. From you results, the rate of sequestration is higher in the leguminous grassland. The hypothesis you propose is that various plant species result in different nutrient conditions in the soil (which ones???), and therefore different decomposition rates. To argue for that, you cite other studies (see Line 205-206) and say that nutrient-poor soils favor faster storage due to slow rates of decomposition…(*this I do not understand: the faster you decompose, the faster you store carbon, right?*) Then you contradict yourself by saying that the storage rate was faster in leguminous grassland because they had greater biomass…this is inconsistent with your above statement about the nutrient effect! The **amount** of stored carbon may be higher because of the greater biomass, but the **rate** of storage shall be linked to the efficiency of the soil to decompose the biomass, right? Maybe you should re-write this section in a clearer way.

**Paragraph 9: Significance of the study and perspective**
Here you HAVE to emphasize the importance of your study! Why is it so important? Can

you give recommendations based on what you observed? What future work should be done? Etc. Put your findings into context!

**7) Comments of specific sections/text**

Section 1: The introduction is too long and part of it should be redirected into the discussion, especially to emphasize the importance of the finding of this study.

Note also that the first objective you describe is "to analyze the effect of SOC stock and sequestration under different grasslands". I do not agree with that: you are not analyzing the **effect of SOC storage**, but the **differences of storage efficiency** under different grasslands, which is very different.

Section 2: In average, the whole section is too long compared to the result/discussion parts, and it can be easily condensed. You are providing a lot of details but you are not using them in the discussion…For example, what is the interest of knowing the cumulative temperature above 10°C and 0°C? How is this important for the present study? You already show the annual temperatures, so I think you don't need this…

Also, is the climate semi-arid or temperate? Can it really be both?

Section 3.2: Do not just refer to you tables but give some key numbers in the text, e.g. by how much does the SOC content increase overtime? L-MS has the highest SOC content: how much?, etc.

Section 3.3: Quantify the increase in terms of difference to make the reading easier, e.g. "the SOC storage under the L-MS, L-CV and L-OV grasslands increased by 5.22, 8.34 and 4.78 Mg.C.ha$^{-1}$ respectively"

Section 3.4:
Line 167: SOC sequestration or SOC sequestration rate?
Line 172-173: The term "respectively" applies to the "maximum and minimum" or to the "L-CV or G-PA grassland"?

**8) Figures**
  1) Add one figure with the studied site localization and perhaps a picture of leguminous and gramineous grassland.
  2) Perhaps add a schematic figure of the sampling strategy, e.g. a vertical profile from above to below the ground with specific location of the samples taken for soil, biomass and other target analyses.

---

## Referee Comment (RC2) · Anonymous Referee #2 · 10 Oct 2016

The authors addressed an interesting topic that the sowing pasture could enhance the ecosystem service for soil carbon sink compared to the abandoned cropland and marginal land in the dry area of northern China. However the way the authors design this experiment can not answer the scientific questions which the authors presented. Also some basic conceptions and definitions for carbon fixation and storage were confused in the manuscripts. For example, SOC sequestration efficiency is commonly expressed by the relationship between annual C input and SOC accumulation rate, however the authors didn't evaluate the difference C input caused by sowing different plant species. Soil bulk density has large variation in the soil depth, the authors didn't describe how to measure and calculate soil bulk density within 100cm, so did SOC

concentration in table 2, is this an average value of 100cm soil depth or something else?

Experiment design and measurement in section 2 is not clear. Many presentation mistakes in the manuscript, line 42-43 (reference sequence), line 47, 53-54, 82.

---

## Short Comment (SC1) · 11 Oct 2016

Topic of carbon sequestration became more and more important for stabilization of postantropogenic ecosystems and climate issues regulation. Carbon sequestration is the one of the most important soil ecosystems services. Nevertheless, manuscript is devoted to assessment of storages of carbon, but not focused on SOM quality. From my point of view the quality of SOM is the most important aspect of stabilization rate assessment. I strongly recommend to discuss it in review, results and discussion as well as in conclusion chapter. This is general comment. The particular ones are follows: a) discussion should be more detailed, with taking into account another types of published data about post antropic successions; b) chemical mechanism of sequestration efficiency should be taken in to account, e.g. characterization of SOM by 13-NMR o by kinetic parameters; c) please provide the soil name in few soil taxonomies, please specify the soil profile morphological organization, name of horizons, soil morphological features, especially of carbonate genesis; d0 please provide more detailed soil chemical and particle size information.
* * *

---

## Referee Comment (RC3) · Anonymous Referee #3 · 26 Oct 2016

Dear Editor, I have finished the review of submitted your journal titled "Cultivated grasslands present a higher soil organic carbon sequestration efficiency under leguminous than under gramineous species" my opinion on the article is as follows: General evaluation Overall evaluation although the content of the study is relatively simple, the results were evaluated well and explanations regarding to findings successfully discussed recent literatures. The subject is the general scope of the journal also; topic is suitable for scope of Solid Earth Abstract. Abstract is informative. Properly keywords are given. The objectives of the article adequate and appropriate in view of the subject matter. Comments and suggestions for improvements Specific Concerns Introduction It is suitable and original contribution. Clearly is reflecting the contents and is relevant with

subject. Material and methods Materials and methods used are suitable and scientific for this article. The description of materials and methods are to allow replication of the experiment. Results are clearly presented. Article structured is in agreement with the Guide for authors. Content justify the length. Tables are all necessary, complete and clearly presented. References are adequate. Conclusions Results and discussion are given clearly. Legends of figures need to be changed. They are currently difficultly understood. Instead of color, different characters can be used in bars Interpretations and conclusions are, justified by the data and consistent with the objectives. This is original contributes to science. Results are clearly expressed. Final Decision The paper deals with an interesting aspect and presents a wide dataset. This study is still able to add some knowledge of interest to an environmental readership. In addition, the paper contains some useful information that is worthy of publication and usefulness for other researchers in this field. In brief, my opinion is, this manuscript has been written in standard scientific way and is suitable for publication in an international journal as like Journal of Solid Earth Acceptable with minor revision, not requiring reconsideration by referee.

---

## Referee Comment (RC4) · Anonymous Referee #4 · 8 Nov 2016

This paper contains the results of an application project that was conducted to determine the most suitable grassland for carbon storage in semi-arid condition. It can be said that this paper is appropriate when it is evaluated as project result report. However, due to the reasons listed below, this paper needs to be improved.

1. The experimental design was not clearly explained (factorial design, randomized block design, randomized plot design etc).

2. The other factors must be constant so that the differences that can occur in dependent variables can be explained only by independent variables. The primary independent variable in this study is the grazing area. The other factors such as soil properties (soil depth, grain size distribution, aggregation), seeding rate, amount of irrigation may

affect the soil carbon content were ignored. Thus this factors must be constant so that changes in soil carbon parameters can only be attributed to the grazing area.

3. The variation in climatic parameters over the years should also be noted.

4. Differences in the grazing area that are considered to be factors were not clearly explained. What is the mean of uncultivated and natural grassland? I don't understand the differences between them.

5. Why the seeding rates were different? How these rates determined?

6. The differences between abandoned cropland and natural grassland in terms of plant covering rates were not given that they can impact the studied carbon parameters.

7. How the bulk density that has been used in the relative calculation formula was measured is not clarified. Where this value was measured in soil profile? In one point or along the profile? As it is well known that soil bulk density can vary along the profile depending on differences in soil properties.

8. Title is not suit for this manuscript. Only two grasslands (leguminous and gramineous grasslands) have been mentioned in the title. However, there are 4 types of grazing compared.

9. The mistake made at the title of the article was also done in the abstract, only the findings of the comparison of the leguminous and gramineous grasslands were given in the summary section.

10. The map showing the study area and sampling points were missing.

11. The descriptive statistics and normality test results of studied properties should be given with a table.

12. The basic soil properties such as grain size distribution, aggregation, pH etc of the grazing areas were not given.

13. When the results are given, it should be indicated in the text that whether the differences are statistically significant or not.

---

## Author Comment (AC1) · 11 Nov 2016

Thank you very much for your positive and constructive comments and suggestions on our manuscript. We have tried to take these comments and suggestions seriously and addressed each of them in all details. We have replied to the comments point by point and all changes have been included in the MS-modified version attached as a supplement.

The scope of the study and the main result are somehow clear, but the writing, however, is rather complicated and in parts lengthy...I strongly suggest a thorough rewriting of the paper to achieve a more logical organization of the discussion, the removal of repetitions and the synthesis of lengthy parts such as the introduction, the presentation of the studied area and the presentation of the data.

Response: Thank you for your suggestions, we have rewritten our manuscript (removed repetitions, restructured the introduction, condensed the presentation of the studied area and Results) to achieve a more logical organization of our paper. The details have been showed in the MS-modified version.

More critical is the lack of rigor of the discussion. Many shortcuts and omissions renders the discussion very difficult to follow, and sometimes borderline when it comes to attribute a process to the observed data...the authors have to discuss more deeply their ideas. Another key problem is that the article fails to attract attention on the importance of such work. Emphasis should be put on the importance of such contribution, not only in the introduction but also through the discussion itself: why does it matter? what's next? what are the perspectives? what broader applicability? etc. Without such perspective, it is hard to evaluate whether the study is of true interest or not for the topic.

Response: Thank you for your suggestion, we have followed your suggestions to reorganize the discussion in the MS-modified version. And the English writing of the following manuscript was carefully edited by a native English speaker (Dr. David Warrington).

Many repetitions can be avoided thorough the entire manuscript: rephrasing the same idea is not useful for the comprehension; it just adds length to the manuscript.

Response: Thank you for your suggestion, we have deleted the repeated sentences through the entire manuscript in our MS-modified version, which has been attached as a supplement.

Please also note the supplement to this comment:
http://www.solid-earth-discuss.net/se-2016-109/se-2016-109-AC1-supplement.pdf

[Figure]

**Supplement:**

**MS-modified version**

**Cultivated grasslands present a higher soil organic carbon sequestration efficiency under leguminous**

**than under gramineous species**

Yu Liu[a,b], Fu-Ping Tian[c], Peng-Yan Jia[a,b], Jing-Ge Zhang[a,b], Gao-Lin Wu[a,b]*

[a] State Key Laboratory of Soil Erosion and Dryland Farming on the Loess Plateau, Institute of Soil and Water

Conservation, Chinese Academy of Sciences and Ministry of Water Resources, Yangling Shaanxi 712100, China;

[b] Institute of Soil and Water Conservation, Northwest A&F University, Yangling Shaanxi 712100, China;

[c] The Lanzhou Scientific Observation and Experiment Field Station of Ministry of Agriculture for Ecological System in the Loess

Plateau Area, Lanzhou Institute of Animal and Veterinary Pharmaceutics Sciences, Chinese Academy of Agricultural Sciences,

Lanzhou, Gansu 730050 China;

* Corresponding author e-mail address: gaolinwu@gmail.com (G.L. Wu)

**ABSTRACT**

The establishment of grasslands on abandoned cropland has been proposed as an effective method of to mitigating climate change by increasing soil organic carbon (SOC) storage. In this study, five cultivated grasslands were established (three leguminous species *Coronilla varia, Onobrychis viciaefolia, Medicago*

*sativa,* and two gramineous species *Poa annua, Agropyron cristatum*), one uncultivated, and one natural grassland were studied to examine how the SOC storage, sequestration rate and sequestration efficiency to change for 5 years restoration in semi-arid area. Our results showed that the cultivated leguminous grasslands had greater total biomass, SOC storage, SOC sequestration rate and efficiency than gramineous grasslands, abandoned cropland, and natural grassland . The greater soil carbon (C) accumulation in leguminous grassland was mainly attributed to the capacity to incorporating carbon and the higher biomass production. Leguminous grasslands accumulated more SOC than gramineous grasslands by 0.64 Mg C ha$^{-1}$

yr$^{-1}$. The average SOC sequestration efficiency in leguminous grassland (1.00) was about 2 times greater than gramineous grassland (0.34). Root nodules in leguminous promote the symbiosis with micro-organisms , which are responsible for the decomposition of the plants, and therefore constitute the key of the transmission of the stored carbon into the soil. The results indicate that cultivated leguminous grasslands sequestered more SOC with higher SOC sequestration efficiency than cultivated gramineous grasslands in arid and semi-arid areas. Our results provide a reference for ecological management in arid and semi-arid areas.

**KEY-WORDS:** Cultivated grassland, Carbon sequestration, Gramineous, Leguminous, SOC

**1 Introduction**

Soil is a key component of the Earth System and it contributes to services, goods, and resources to the humankind (Brevik et al., 2015). Soil  stores more carbon (C) than the atmosphere and vegetation (Köchy et al., 2015; Keesstra et al., 2016). Soil organic carbon (SOC) as a key component of the global carbon cycle and its potential to sink from atmosphere carbon dioxide ($CO_2$) have been widely discussed in the scientific literatures throughout the world (Guo and Gifford, 2002; Lal, 2004; De Deyn et al., 2008; Deng et al., 2014a; Parras-Alcántara et al., 2015).  In the terrestrial ecosystem, SOC pool dynamics were affected by many factors, such as climate change (Lal, 2004; Field et al., 2007), management practices (Luo et al., 2010; Ono et al., 2015), land use etc. (Post et al., 2000; Don et al., 2011; Deng et al., 2014b; Muñoz-Rojas et al., 2015).

SOC plays an extremely important role in controlling soil fertility and cropping system productivity and sustainability (Hurisso et al., 2013; De Moraes Sá et al., 2015), particularly in low-productivity arid and semiarid agro-ecosystems (Behera et al., 2015). To develop farming methods that  conserving SOC is therefore of a great importance (Lal, 2004). Cultivated grassland has much more advantages than natural grassland regeneration, such as accelerating vegetation restoration and improving grassland productivity. Establishing artificial grassland is one type of land uses to  restoring vegetation and  improving SOC (Fu et al., 2010; Wu et al., 2010; Li et al., 2014).

respiration from plant roots (Amundson, 2001; Garcia-Diaz et al., 2016). The biomass fraction resulting in

SOC build-up (plant residuals) was strongly affected by management practices including the selection of plant species (Don et al., 2011). Species composition had a great role in determining the aboveground productivity (Liu et al., 2016). Over relatively long time, the proportion of the aboveground biomass enters soil as organic matter and incorporates into soil through physical and biological processes. For example, some leachates from plant material in the litter layer, root exudates, solid decomposed litter and fragmented plant structure materials (Jones and Donnelly, 2004; Novara et al., 2015). The amount of plant residuals returned to the soil directly affected the SOC (Musinguzi et al., 2015; Wasak et al., 2015), and mostly perennial plants were managed with high planting densities to produce greater biomass exports (Hobbie et al.,

2007; Köchy et al., 2015).

Vegetation degradation and exponential population growth have caused massive amounts of soil and water to be lost. The Chinese government has implemented the most ambitious ecological program titled

'Grain-for-Green' Project (converting degraded, marginal land and cropland into grassland, shrubland and forest), with the objective of transforming the low-yield slope cropland into grassland, reducing soil erosion, maintaining land productivity and improving environmental quality (Fu, 1989; Liu et al., 2008;). The large scale of the project indeed enhanceed carbon sequestration capacity in China, especially in arid and semi-arid areas (Chang et al., 2011; Song et al., 2014).

Many prior studies about SOC have paid much attention to convertingsion from farmland to grassland, shrubland or forest (Fu et al., 2010; Deng et al., 2014a). The main dominant grass species used in the project are leguminous and gramineous (Jia et al., 2012; Wang et al., 2015). However, less attention has been devoted to the SOC among different plant species grasslands. In current study, we have focused on ascertaining the influence of leguminous and gramineous grasslands on SOC sequestration capacity and efficiency. Many studies had demonstrated that there is a significant and positive relationship on SOC and nitrogen (Deng et al., 2013; Zhu et al., 2014). So we hypothesize that the leguminous grassland has the higher SOC sequestration capacity than gramineous grassland. More specifically, our objectives are: (i) to analyze the differences of storage efficiency under different grasslands; (ii) to determine which type of cultivated grassland might better improve SOC storage in arid and semi-arid areas.

**2 Material and methods**

**2.1 Experimental site and design**

The study   was conducted at the Lanzhou scientific observation and experiment field station of ministry of agriculture for ecological system in the Loess Plateau area (103°44.342′ E, 36°02.196′ N, 1635 m a.s.l.) in Lanzhou, Gansu Province, China (Fig. 1). The site is the  temperate continental  climate zone. The data from the National

Meteorological Information Center of China showed that the mean annual temperature is 9.3 ℃

and the minimum and maximum values were -23.1 ℃ and 39.1 ℃ (2008-2012), respectively.

The topography of study area is typical characteristics of the

Loess Plateau, such as plains, ridges and mounds, etc.  Soil parent material is the Quaternary eolian loess. The main soil type is Sierozem, which is a calcareous soil and characteristics of the Chinese loess region (Li et al., 2010). Sierozem is the soil developed in the dry climate and desert steppe in warm temperate zone, which has low humus and weak leaching (National soil census office, 1998). There is the patch or pseudohyphae calcium carbonate deposition and strong lime reaction within full sierozem profile (Shi, 2013). Soil total nitrogen content is about 0.41g $kg^{-1}$, total phosphorus content is 0.46 g $kg^{-1}$, total potassium content is 18.24 g $kg^{-1}$, and pH is 8.25 in study site.

The experimental site was originally under sorghum (*Sorghum bicolor* L.) continuously from 1970 to 2005 and it was abandoned from 2005 to 2007 (grazing exclusion). In 2007, five cultivated grasslands, one uncultivated grassland (abandoned cropland, Un-G), one natural grassland (Na-G) were established in the study site. Five main forage grasses, widely grown across in semi-arid areas, were selected to establish five types of cultivated grassland, namely three leguminous species (*Coronilla varia* L., L-CV; *Onobrychis viciaefolia* Scop, L-OV; *Medicago sativa* L., L-MS) and two gramineous species (*Poa annua* L., G-PA; *Agropyron cristatum* L. Gaertn., G-AC) (Table 1). The seeding rates in different grassland were showed in Table 1. The different seeding rates were contributed to the percentage of germination, to guarantee the equal plant density in each grassland. The rates were determined by reference to local farmland crop density. We designed the experiment to be a randomized plot design. Three experimental plots (10 m × 20 m) were established randomly within each of the grassland areas. The forage grasses were planted in early April of 2007, and all plots were weeded manually and watered three times (April, June, October) annually from 2008 to 2012 to preserve the monocultures. The plots did not fertilized during cultivation. All the plots were harvested once a year in October.

**2.2 Aboveground plant and belowground biomass sampling**

Ten quadrats (1 m × 1 m) were randomly arranged in each plot in late August every year (2008-2012). Aboveground biomass was measured by harvesting the upper plant parts, by (clipping their stems at the soil surface), from ten each quadrats (1 m × 1 m) in each plot randomly in late August every year (2008-2012). All green aboveground plant parts in each quadrat were collected separately by each individual species, and all the litters layer also were collected with the labeled envelops. Then these samples were dried at 105 ℃ until their mass was constant, and then their mass was weighed and recorded.

Belowground biomasses and soil samples were taken in the four corners and the center of the each quadrats where were the aboveground biomass sampling points is (Fig. 1). Belowground biomass were was collected using a soil drilling sampler with 9 cm inner diameter  at 0-100 cm soil layer ( separated into increments every 10 cm). The roots in the soil samples were obtained by a 2 mm sieve. Then the remaining roots in the soil samples were isolated by shallow trays, and allowing the flowing water from the trays to pass through a 0.5 mm mesh sieve. All the roots samples were oven-dried at 65 ℃ then weighed.

**2.3 Soil sampling and determination**

The method of soil sampling is similar to belowground (using a soil drilling sampler with 4 cm inner diameter from 0-100 cm soil layer). In each quadrat, the same layer samples (every 10 cm) were mixed together to composed  a composite sample. The samples were passed through a 2-mm sieve to remove the roots and other debris. A 5 cm diameter and 5 cm high stainless steel cutting ring ($\sim$100 cm$^3$) was used to measure soil bulk density (BD) at adjacent points to the soil sampling . Soil bulk density was measured at the depth of 0-100 cm (10 cm a layer then averaging). The dry mass  was measured after oven-drying at 105 ℃. Soil organic carbon content was measured using the method of the vitriol acid-potassium dichromate oxidation (Walkley and Black, 1934). All the analyses of one sample were carried out in three replications.

**2.4 Relative calculation**

BD was calculated depending on the oven dried weight of the composite soil samples (Deng et al., 2013).

The SOC stock for each soil layer was calculated using the equation as follows (Deng et al., 2013):

$$C_s = BD \times SOC \times D/10 \tag{1}$$

where, $C_s$ is the SOC stock (Mg ha$^{-1}$); BD is the soil bulk density (g cm$^{-3}$); SOC is the soil organic carbon content (g kg$^{-1}$); and D is the thickness of the sampled soil layer (cm).

The SOC sequestration rate (SSR, Mg ha$^{-1}$ yr$^{-1}$) was calculated as follows (Hua et al., 2014):

$$SSR = (C_t - C_0)/t \tag{2}$$

where, $(C_t - C_0)$ is SOC sequestration; $C_t$ is the SOC stock in 2012; $C_0$ is the SOC stock in 2008; t was the duration of experiment (year).

The SOC sequestration efficiency was estimated using the SOC sequestration in the weight of total biomass (aboveground biomass and belowground biomass) of per unit area:

$C_{se} = \triangle C / B_T / 10$                                                      (3)

where, $C_{se}$ is the SOC sequestration efficiency; $\triangle C$ (Mg ha$^{-1}$) is the SOC sequestration from 2008 to

2012; $B_T$ (kg m$^{-2}$) is the total biomass (above ground and below ground) from 2008 to 2012.

**2.5 Statistical analyses**

The data were examined for normality by the Shapiro-Wilk test and homogeneity of variances by the Levene test before analysis (Table 2). To get a normal distribution, performing statistical tests not normally distributed data were log-transformed. All data were expressed as mean values ± standard error (M ± SE).

The means of SOC sequestration rate and SOC sequestration efficiency among the different grassland types were assessed using One-way Analysis of Variance (ANOVA). Two-way ANOVA of Type III was performed to test the influences of grassland types and time on SOC content, storage and bulk density. Tukey test was conducted to test the significance at $p < 0.05$ level. All the statistical analysis was performed with SPSS

version 18.0 (SPSS Inc., Chicago, IL, USA).

**3 Results**

**3.1 Aboveground net primary productivity**

Between 2008 and 2012, the five cultivated grasslands in general had greater total biomass values than the uncultivated grassland and natural grassland (mean by 189.36%). In addition, the three grasslands cultivated with the leguminous species had greater annual total biomass than the two gramineous grasslands (mean by

72.6%), which lead to a greater total biomass values of the three leguminous species at the end of the study period. In particular, the L-MS grassland consistently had the greatest total biomass throughout the study period (Fig. 1a2a).
* * *
Results from two-way ANOVA showed that the plots, year and interactions all significantly affected total biomass, SOC content, and BD ($p < 0.001$, Table 3). The average SOC content followed leguminous grassland $(4.21 \pm 0.31 \text{ g kg}^{-1})$ > natural grassland $(2.90 \pm 0.14 \text{ g kg}^{-1})$ > uncultivated grassland $(2.58 \pm 0.17$ $\text{g kg}^{-1})$ > gramineous grassland $(2.46 \pm 0.15 \text{ g kg}^{-1})$, and it increased over time in all grasslands (Table 4).  The effects of grassland type on soil bulk density followed uncultivated and natural grassland $(1.44 \pm 0.02 \text{ g cm}^{-3})$ > gramineous grassland $(1.43 \pm 0.01 \text{ g cm}^{-3})$ > leguminous grassland $(1.40 \pm 0.01 \text{ g cm}^{-3}$, (Table 5).
* * *
The SOC storage under all the grasslands increased significantly throughout the study period (Table 6))   In the 0-20 cm soil layer, the SOC storage under the L-MS, L-CV and L-OV grasslands increased by 5.22, 8.34 and 4.78 Mg C ha$^{-1}$, respectively, during the experimental period.
* * *
 Three leguminous grasslands accumulated C with an average rate 1.00 Mg C ha$^{-1}$ yr$^{-1}$ which is more than the 0.34 Mg C ha$^{-1}$ yr$^{-1}$ in gramineous grassland, and more than the average of uncultivated and natural grasslands (0.25 Mg C ha$^{-1}$ yr$^{-1}$; Fig.2c).

The mean SOC sequestration efficiency in the leguminous grassland was about 0.26, which was significantly greater than others grassland types (0.13; $p$<0.05; Fig.2d).

**4 Discussion**

   Grassland has a significant effect in arid and semi-arid areas carbon cycle through changing soil carbon accumulation rates and turnover, soil erosion, and vegetation biomass (Deng et al.,2014a; Liu et al., 2016a). Plant regulated SOC stock by controlling carbon assimilation, transfer and storage in the plant root system, then through plant respiration and leaching release from soil to atmosphere(De Deyn et al., 2008; Garcia-Diaz et al., 2016). SOC inputs mostly originate from decaying aboveground and belowground plant tissue, so greater soil C accumulation was mainly ascribed to increasing soil C input from higher biomass production (Deng et al., 2014c; Wu et al., 2016). Mutualistic symbionts (N-fixing bacteria and mycorrhizal fungi) are also an important source of carbon input to soil, especially in actively growing plants (Bardgett et al., 2005). In grassland, atmosphere carbon  is sequestrated through photosynthesis and respiration, then carbon  fixes in stable SOC pool or  release back into the atmosphere (Post et al., 2000). Therefore, studying the carbon sequestration in grassland ecosystems can help to identify the magnitude of global carbon sinks and sources (Li et al., 2014). Our results showed that leguminous grassland had greater SOC content and storage efficiency than gramineous grassland. ~~SOC content of all grassland plots showed some differences between each other (Table 2 and 3). The average SOC content in leguminous grasslands was 2.64 g kg$^{-1}$ and that in gramineous grasslands was 1.97 g kg$^{-1}$. Moreover, both soil bulk density of leguminous and gramineous grasslands were 1.46 g cm$^{-3}$ in 2008. The reasons for the SOC content difference result from precedent soil conditions and cultivated grasses.Dt types, as well as the precedent soil conditions aretwosSOC content~~ differences between leguminous and gramineous grasslands.

   Different species may incorporate more or less carbon according to their specific metabolism. Legumes have been identified as a key driver of C sequestration in many studies (Fornara and Tilman, 2008; Wu et al., 2016).

soil and changing the soil physical conditions, such as SOC and BD. In addition, mutualistic symbionts (N-fixing bacteria and mycorrhizal fungi) are also an important source of carbon input to soil, especially in actively growing plants (Bardgett et al., 2005). Symbiosis can increase plant productivity through enhanced the acquisition of limited resources. Legumes live in a symbiosis with *Rhizobium* bacteria, which fix atmospheric N. Moreover, many previous studies had demonstrated that soil carbon and total nitrogen are significantly and positively correlated (Deng et al., 2013; De Oliveira et al., 2015). Therefore, mycorrhizal fungi can immobilize carbon in their mycelium and improve carbon sequestration in soil aggregates (Rillig and Mummey, 2006). It might be expected that the cultivated leguminous grasslands significantly improved soil N content that led to the greater carbon sequestration ability than the non-leguminous grasslands. Moreover, mycorrhizal fungi can immobilize carbon in their mycelium and improve carbon sequestration in soil aggregates (Rillig and Mummey, 2006). Our results demonstrated that a key variable associated with higher SOC content in leguminous grasslands than gramineous grasslands is the greater total biomass accumulation. The leguminous grasslands had both higher above- and belowground biomasses than gramineous grasslands. Total biomass was 16.35 kg m$^{-2}$ in leguminous grasslands, which is 9.47 kg m$^{-2}$ more than gramineous grasslands from 2008 to 2012. In addition, the grasslands in our study without grazing and only harvesting the aboveground biomass annually, so all the aboveground stubble and plant litters be input to soil as a carbon supply. SOC mostly originates from decaying this aboveground and belowground plant tissue, so greater soil C accumulation was mainly ascribed to increasing soil C input from higher biomass production (Deng et al., 2014c; Wu et al., 2016). Previous studies had showed that plant regulated SOC stock by controlling carbon assimilation, its transfer and storage in plant root system, then through plant respiration and leaching its release from soil to atmosphere(De Deyn et al., 2008). In addition, the biomass fraction resulting in SOC build-up (plant residuals) was strongly affected by management practices including the selection of plant species (Don et al., 2011). Species composition had a great role in determining the aboveground productivity (Liu et al., 2016b). Over relatively long time, the proportion of the aboveground biomass enters soil as organic matter and incorporates into soil through physical and biological processes.

Some leachates from plant material in the litter layer, root exudates, solid decomposed litter and fragmented plant structure materials were the main sources of soil organic matter (Jones and Donnelly, 2004; Novara et al., 2015). The amount of plant residuals returned to the soil directly affected the SOC (Musinguzi et al.,

2015; Wasak et al., 2015), and mostly perennial plants were managed with high planting densities to produce greater biomass exports (Hobbie et al., 2007; Köchy et al., 2015). Deng et al. (2014c) have found that plant biomass is the key driver in soil carbon sequestration. In this study, the SOC increased dramatically in leguminous grassland due to the greater total biomasses of the leguminous grasses, which resulting in  the increasing soil carbon inputs from the litter layer and root biomass (De Deyn et al., 2008; Wu et al.,

2010; Novara et al., 2015). Moreover, Symbiosis can increase plant productivity through enhancing the acquisition of limited resources. Our results demonstrated that a key variable associated with higher SOC

content in leguminous grasslands than gramineous grasslands is the greater total biomass accumulation. The leguminous grasslands had both higher above- and belowground biomasses than gramineous grasslands.

Total biomass was 16.35 kg m$^{-2}$ in leguminous grasslands, which are 9.47 kg m$^{-2}$ more than gramineous grasslands from 2008 to 2012. In addition, the grasslands in our study without grazing and only harvested the aboveground biomass annually, so all the aboveground stubble and plant litters input to soil as a carbon supply.

SOC sequestration rates in the  leguminous grasslands were significantly higher than that in the gramineous grasslands (Fig. 1c). This maybe resulted from SOC sequestration and the different decomposition rates in soils, because the  leguminous and gramineous grass species result in multifarious nutrient conditions.

 Litter and fragmented plant parts at the soil surface are decomposed by micro-organisms and are gradually incorporated into the soil through some complex processes, such as humification and mineralization (Novara et al., 2015). Legumes had the ability to develop root nodules and to fix nitrogen in symbiosis with compatible rhizobia, which should improve the soil nutrient status and microbial community. In addition to this, rates of decomposition in leguminous grassland may be higher due to excellent soil physical and chemical conditions. Root nodules promote the symbiosis with micro-organisms, which are responsible for the decomposition of the plants, and therefore constitute the key of the transmission of the stored carbon into the soil.  Furthermore, the  increasing  fertility of the soils in  the leguminous grasses should facilitate the  increasing productivity of the plants. Our results showed that SOC sequestration efficiency under leguminous grassland was evidently greater than that in the gramineous grassland (Fig. 1d). It is noteworthy that L-MS grassland had the highest total biomass 22.59 kg m$^{-2}$ which is 2.38 times as much as the average of gramineous grasslands (Fig. 2a Moreover, SOC sequestration in L-MS grassland is 3 times as much as the average of gramineous grasslands (Fig. 2b). The differences of species biological characteristics determine the capacity of carbon sequestration. So the SOC sequestration efficiency in L-MS grassland is higher than gramineous grassland.

Despite the indications from this study of higher SOC sequestration rate and efficiency in leguminous grassland, specific researches are still needed to determine the potential mechanisms of each species in sequestrating carbon. Many studies have been demonstrated legumes are higher water consuming plants than gramineous in arid and semi-arid areas, so it is necessary to balance the ecological effect of grassland for rational utilization of resources.

**5 Conclusion**

Leguminous grasslands had greater SOC storage, sequestration rate and efficiency than gramineous grasslands. The greater soil C accumulation of leguminous grasslands was mainly ascribed to the capacity to incorporate carbon and the higher biomass production. Leguminous grasslands accumulated an average rate of 0.64 Mg C·ha$^{-1}$·yr$^{-1}$ more than gramineous grasslands. The average SOC sequestration efficiency in leguminous grasslands was 2 times greater than that in the gramineous grasslands. The Our results indicate that cultivated leguminous grasslands sequestered more soil carbon with a higher SOC sequestration efficiency than cultivated gramineous grasslands in arid and semi-arid areas. Our results provide a reference for ecological management in arid and semi-arid areas.

**6 Acknowledgement**

We thank the editor for suggestions on this manuscript. This research was funded by the National Natural Science Foundation of China (41371282, 41525003, 31372368, 41371282, and 41303062), the Youth Innovation Promotion Association CAS (2011288), the "Light of West China" Program of CAS (XAB2015A04), Lanzhou Institute of Animal and Veterinary Pharmaceutics Sciences of Chinese Academy of Agricultural Sciences (CAAS-ASTIP-2014-LIHPS-08).

[revised manuscript text omitted]

**Table** 3 The *p* values of homogeneity of variances by the Levene test and normality by the Shapiro-Wilk test in soil organic carbon content (SOC), soil C storage, and soil bulk density (BD).

[revised manuscript text omitted]

level. The dotted lines indicate the means of the same grassland types.

---

## Author Comment (AC3) · 11 Nov 2016

Thank you very much for your positive and constructive comments and suggestions on our manuscript. We have tried to take these comments and suggestions seriously and addressed each of them in all details. We have replied to the comments point by point and all changes have been included in the MS-modified version attached as a supplement.

SOC sequestration efficiency is commonly expressed by the relationship between annual C input and SOC accumulation rate, however the authors didn't evaluate the difference C input caused by sowing different plant species. Soil bulk density has large variation in the soil depth, the authors didn't describe how to measure and calculate

soil bulk density within 100cm, so did SOC concentration in table 2, is this an average value of 100cm soil depth or something else?

Response: Thank you for your suggestion, the SOC sequestration efficiency was used to calculate the amount of carbon sequestration on the aboveground biomass of per unit area, which was put forward by us. We have added the description of soil bulk density and SOC concentration in table 4 and 5, which is an average value of 100 cm soil depth. The details have been showed in the MS-modified version.

Experiment design and measurement in section 2 is not clear. Many presentation mistakes in the manuscript, line 42-43 (reference sequence), line 47, 53-54, 82.

Response: Thank you for your suggestion, we have revised our experiment design and measurement in section 2 and other mistakes through the entire manuscript to make it clear and rigorous. Moreover, we have added a figure to show our study site and sampling strategy in modified version Figure 1.

Please also note the supplement to this comment:
http://www.solid-earth-discuss.net/se-2016-109/se-2016-109-AC3-supplement.pdf
* * *

---

## Author Comment (AC4) · 11 Nov 2016

Thank you very much for your positive and constructive comments and suggestions on our manuscript. We have tried to take these comments and suggestions seriously and addressed each of them in all details. We have replied to the comments point by point and all changes have been included in the MS-modified version attached as a supplement.

Manuscript is devoted to assessment of storages of carbon, but not focused on SOM quality. From my point of view the quality of SOM is the most important aspect of stabilization rate assessment. I strongly recommend to discuss it in review, results and discussion as well as in conclusion chapter. This is general comment. The particular

ones are follows: a) discussion should be more detailed, with taking into account another types of published data about post antropic successions; b) chemical mechanism of sequestration efficiency should be taken in to account, e.g. characterization of SOM by 13-NMR o by kinetic parameters; c) please provide the soil name in few soil taxonomies, please specify the soil profile morphological organization, name of horizons, soil morphological features, especially of carbonate genesis; d0 please provide more detailed soil chemical and particle size information.

Response: Thank you very much for your positive and constructive comments and suggestions on our manuscript entitled "Thank you very much for your positive and constructive comments and suggestions on our manuscript entitled". In our study we calculated SOC by the relationship between SOM and SOC (Guo &Gifford, 2002) using the formula: SOC=0.58×SOM (Deng et al., 2014), so SOM and SOC have the same changing tendency in our study. Moreover, we have added some discussion about SOM in our manuscript. For example, "Over relatively long time, the proportion of the aboveground biomass enters soil as organic matter and incorporates into soil through physical and biological processes. Some leachates from plant material in the litter layer, root exudates, solid decomposed litter and fragmented plant structure materials were the main sources of soil organic matter." We have revised our discussion by re-organizing the structure and adding some details. The soil properties have been added in section 2. The details and all the changes have been included in the MS-modified version attached as a supplement.

Please also note the supplement to this comment:
http://www.solid-earth-discuss.net/se-2016-109/se-2016-109-AC4-supplement.pdf

---

## Author Comment (AC6) · 11 Nov 2016

Thank you very much for your positive and constructive comments and suggestions on our manuscript. We have tried to take these comments and suggestions seriously and addressed each of them in all details. We have replied to the comments point by point and all changes have been included in the MS-modified version attached as a supplement.

The experimental design was not clearly explained (factorial design, randomized block design, randomized plot design etc).

Response: Thank you for your suggestion, we have added the description of exper-

imental design in the section 2, and we have clearly explained that we designed the experiment to be a randomized plot design in our modified version.

The other factors must be constant so that the differences that can occur in dependent variables can be explained only by independent variables. The primary independent variable in this study is the grazing area. The other factors such as soil properties (soil depth, grain size distribution, aggregation), seeding rate, amount of irrigation may affect the soil carbon content were ignored. Thus this factor must be constant so that changes in soil carbon parameters can only be attributed to the grazing area.

Response: Thank you for your suggestion, the experimental site was originally under sorghum (Sorghum bicolor L.) continuously from 1970 to 2005 and it was abandoned from 2005 to 2007 (grazing exclusion). We conducted our experiment in 2007 and it was without grazing from 1970 to 2012, so we studied the effect of plant species on soil carbon.

The variation in climatic parameters over the years should also be noted.

Response: Thank you for your suggestion, we had compared the average of precipitation and temperature in this site before we analyzed the data. There was no significant difference from 2007 to 2012.

Differences in the grazing area that are considered to be factors were not clearly explained. What is the mean of uncultivated and natural grassland? I don't understand the differences between them.

Response: Thank you for your suggestion, uncultivated grassland in our paper represents abandon cropland, which convert from cropland to grassland. It was not ploughed during our study period, so we used uncultivated grassland. Natural grassland represents the grassland never has never been ploughed.

Why the seeding rates were different? How these rates determined?

Response: Thank you for your suggestion, the different seeding rates were contributed

to the percentage of germination, to guarantee the equal plant density in each grassland. The rates were determined by reference to local farmland crop density.

The differences between abandoned cropland and natural grassland in terms of plant covering rates were not given that they can impact the studied carbon parameters.

Response: Thank you for your suggestion, according to our investigation, the plant covering rates of abandoned cropland and natural grassland were no significant difference, so those were not showed in our manuscript.

How the bulk density that has been used in the relative calculation formula was measured is not clarified. Where this value was measured in soil profile? In one point or along the profile? As it is well known that soil bulk density can vary along the profile depending on differences in soil properties.

Response: Thank you for your suggestion, we calculated bulk density layer by layer (10 cm). We used an average of ten layers (0-100 cm) to show the bulk density in the depth of 0-100 cm.

Title is not suit for this manuscript. Only two grasslands (leguminous and gramineous grasslands) have been mentioned in the title. However, there are 4 types of grazing compared. The mistake made at the title of the article was also done in the abstract, only the findings of the omparison of the leguminous and gramineous grasslands were given in the summary section.

Response: Thank you for your suggestion, title is a brief summary of the main results of the article. Although there are 4 types of grasslands in our manuscript, the main results focus on leguminous and gramineous grasslands and uncultivated grassland and natural grassland are just control in our study.

The map showing the study area and sampling points were missing.

Response: Thank you for your suggestion, we have added a map and a schematic figure of the sampling strategy in Figure 1 to show the study area and sampling points.

The descriptive statistics and normality test results of studied properties should be given with a table.

Response: Thank you for your suggestion, we have added the descriptive statistics and normality test results of studied properties in Table 2.

The basic soil properties such as grain size distribution, aggregation, pH etc of the grazing areas were not given.

Response: Thank you for your suggestion, we have added the relative content in section 2 experimental site and design in our modified version.

When the results are given, it should be indicated in the text that whether the differences are statistically significant or not.

Response: Thank you for your suggestion, we have revised our conclusion and abstract to show the main result and the implications of the results in our modified version.

Please also note the supplement to this comment:
http://www.solid-earth-discuss.net/se-2016-109/se-2016-109-AC6-supplement.pdf